# Preferences for Sweet and Fatty Taste in Children and Their Mothers in Association with Weight Status

**DOI:** 10.3390/ijerph17020538

**Published:** 2020-01-15

**Authors:** Grzegorz Sobek, Edyta Łuszczki, Mariusz Dąbrowski, Katarzyna Dereń, Joanna Baran, Aneta Weres, Artur Mazur

**Affiliations:** Medical College of Rzeszów University, 35-959 Rzeszow, Poland; g.sobek@wp.pl (G.S.); eluszczki@ur.edu.pl (E.Ł.); mariusz.dabrowski58@gmail.com (M.D.); joannabaran.ur@gmail.com (J.B.); anetaweres.ur@gmail.com (A.W.); drmazur@poczta.onet.pl (A.M.)

**Keywords:** dietary patterns, food preference, overweight and obesity, sensory perception

## Abstract

Strong preferences for sweet and fat tastes (characters) are associated with the consumption of foods high in calories. The taste preferences, especially the sweet and fat tastes (characters), might be one of the factors predisposing children to become overweight and/or develop obesity. The aim of the study is to assess taste preferences in children and their mothers in association with their weight status. In the study, 150 children aged 8–15 were included; among them, 75 had overweight and/or obesity, and 150 mothers of whom 69 were overweight and/or obese. Body composition estimates were obtained using a bioelectrical impedance analysis-body height was measured using a stadiometerSeca 213. Sensory tests were carried out using apple juice of various sugar content and crackers of various fat content. Results show that children prefer a sweet taste more often than their mothers (50.0% vs. 35.3%, *p* = 0.009). In the group of children who preferred the high-sweet taste, there were twice as many obese children when compared to the group who preferred the low-sweet taste. Similar relationships applied to mothers. Preferences for fat taste (character) among mothers increased the risk of obesity among their children (39% vs. 20%; *p* = 0.039). Taste preferences, especially a sweet taste preference, seem to be one of the important factors determining overweight and obesity in children and adults. The fat taste(character) preferences in mothers correlate with the overweight and/or obesity of their children, while such preferences among children were not significantly different irrespective of body weight status.

## 1. Introduction

The increasing prevalence of overweight and obesity in children and adults around the world is a critical public health problem [1,2,3,4]. Eating habits and food preferences belong to those environmental factors that significantly increase the risk of overweight and/or obesity [5]. Until the first decade of the 21st century, it was assumed that the sense of taste included 5 main flavors: sweet, salty, sour, bitter, and glutamate [6,7,8]. Studies in the last decade also documented the important role of perception of fat taste (character) in animal and human behavior [9,10,11,12,13]. The ability to perceive flavors begins in the uterus through the amniotic fluid as well as breast milk, along with the development and early functioning of the taste and olfactory system [14]. Tastepreferences are created in early childhood [15]. In general, children prefer sweet and dairy products as well as foods rich in fat, which are characterized by high energy density. In the later period, the taste impression is also determined by the supply of specific food products (eating habits, culture) and experiences [16,17,18]. There is evidence that eating habits learned in early childhood are often continued during adulthood [19]. Adult food preferences are associated with age, sex, health status, education, income [20], and the healthfulness of food preferences increases with increasing age [21]. Experiments with adults studying the relationship between taste preferences and overweight and/or obesity have led to ambiguous conclusions [22,23,24,25,26]; moreover, the methodology was questioned by Bartoshuk et al. [27]. Available studies also confirm that “liking” or a strong preference for children’s sweet taste is associated with a higher intake of foods rich in sugar, sweet beverages, or the preference for high-sugar cereal flakes [28]. Heterogeneous methodology means that positive relationships between preferences of sweet or fat taste (character) and the occurrence of overweight or obesity in children could be only suggested. The research uses the taste preference test method, which, according to Cox et al. [29] (the author of the latest literature review in this field), has, so far, the highest methodological quality.

A few reports are available in the literature analyzing the impact of children’s taste preferences in the context of overweight and/or obesity and adult taste preferences separately. So far, there are no reports in the literature comparing the taste preferences of children and their parents to weight status. Investigating the relationship of mothers’ taste preferences to the weight of children will allow us to perform a more accurate assessment of this element of obesity risk. The main aim of this study is to analyze the relationship between the taste preferences of participants (children, mothers) and overweight and obesity. It is also analyzed whether the mother’s taste preferences determined the children’s taste preferences and the occurence of overweight and obesity.

## 2. Materials and Methods

### 2.1. Participants

The study is a preliminary assessment of the relationship between the taste preferences of mothers and their children in relation to the weight of mothers and children. Two hundred thirty-ninechildren aged 8–15 years were enrolled in the study. Children were recruited from two schools selected via a randomized algorithm, one located in an urban environment (StanisławWyspiański Secondary School Complex No. 3 in Rzeszów, Poland) and another in a rural area (Complex of Schools in Kosina, Poland). In this group, 75 children with overweight or obesity were identified. These children constituted the study group. From the remaining 164 children with normal weight, 75 comparators were selected in a 1:1 ratio, strictly matched by age (the nearest date of birth) and gender to the study group. The selection of study participants is presented in Figure 1. There were also 150 mothers of the participants included in the study. Mothers were chosen because of their impact on the development of childhood eating habits. Sixty-nineof them had overweight and/or obesity (BMI ≥25 kg/m^2^) [30]. The inclusion criteria for children were age 8–15, the attendance of one of two selected schools, and parents’ acceptance to participate in the study. The requirement for the child’s participation in the study was the simultaneous participation of his/her mother. The exclusion criteria for children included suffering from chronic diseases affecting body weight, underweight (<5th percentile), inability to consume food samples used in the study, implanted pacemaker and pregnancy (contraindications for bioimpedance testing). The children participating in the study were a representative sample of the population. The inclusion and exclusion criteria for mothers were the same, except for the age and body weight. During the parents meeting with teachers at schools, the main goals of the study were presented. Attending mothers were asked to participate in the study by themselves and to permit their children to participate. It was noted that the research is voluntary and has only a scientific purpose. The study was conducted after obtaining written consent from the participating children’s parents and the children themselves. All participants and parents were fully informed in writing and verbally about the nature of the study.

### 2.2. Assessments

The study was conducted according to the method used to assess taste preferences (sweet taste, fat taste) in the IDEFICS research [31]. Sensory tests were carried out in the food sensory laboratory in the Centre for Innovative Research in Medical and Natural Sciences (University of Rzeszów, Rzeszów, Poland), where suitable conditions for conducting sensory evaluation was guaranteed. It was ensured that there were no unwanted odors, such as strong smells of food or disinfectants. The room was free of sources of distraction. To assess the taste preference for sweet and fatty tastes, forced-choice paired preference tests with apple juice (sweet) and with crackers (fatty) were applied, as described in detail by Knof [32]. Samples of apple juice and crackers, identically prepared as in the method described by Knof, were used in the study. The samples had been obtained from the leader of the IDEFICS consortium—Leibniz Institute for Prevention Research and Epidemiology-BIPS—prepared for subsequent sensory tests as part of the “I Family” study (funded by the EC FP7 Project No 266044). Each child/mother had to choose a preferred sample out of a pair. The pairs consisted of two samples, one was the basic sample and the other one the modified sample. Sweet preference was measured with a pair of apple juices. For the sweet preference tests, clear apple juice was offered. The basic sample contained 0.53% added sucrose and the high-sugar sample contained 3.11% added sucrose. The juices were served in small cups with a volume of 30 mLat 18±2 °C. Fat preference was assessed using a cracker as a food sample. The reference cracker was prepared from the basic recipe that consists of water, flour (wheat), fat (8%), and salt. For the modification, the amount of fat was increased to 18%. All crackers were heart-shaped, were coated with 0.5% aqueous solution of soda lye and prepared in an industrial oven. Baking conditions were 2 min at 300 °C, followed by 5 min at 200 °C. For the crackers with high-fat content, the main baking time was extended to 5 min and 30 s to ensure the consistent color of all crackers. High-fat preference was assumed when the child chose the cracker with the added fat over the basic cracker. Crackers were sealed in plastic bags and juices were sterilized and packed into tetra packs. The outcome variables were coded with 1 or 2 depending on the preferred sample of the test pairs.

Children’s anthropometric measurements were carried out at the Centre for Innovative Research in Medical and Natural Sciences (Rzeszow, Poland). All measurements were taken between 7:00 and 10:00 am by experienced researchers.

Height measurements were made three times with a SECA 213 portable stadiometer, with an accuracy of 5 mm, in a standing position, upright and without footwear. The average figure of the three measurements was used in the analyses. Body weight was assessed with an accuracy of 0.1 kg using a body composition analyzer (BC-420, Tanita, Tokyo, Japan). According to the instructions for the Tanita BC 420 device for accurate measurement, the machine was positioned as horizontally as possible. The adjustable feet had been rotated in 4 positions to adjust so that the level indicator bubbles get inside. Participants stood on the platform barefoot, upright, on straight legs and made sure that the front of the feet touched the front electrodes and rear parts of the rear electrodes. During the measurement, the participant should be stationary and the bodyweight evenly distributed between both feet. The height and weight of all participants were measured in fasting status wearing underwear. Body mass index (BMI) was calculated as weight (kg)/height (m)^2^ based on BMI values; the BMI percentile of individual participants was calculated. BMI percentile charts specific for age, sex, and body height were used. Percentile charts that were developed within the framework of the Polish project entitled “Developing standards of blood pressure in children and adolescents in Poland, OLAF” were used [33]. Based on the BMI percentile values, underweight (<5th percentile), normal weight (between 5th and 85th percentile), overweight (BMI ≥85th percentile and <95th percentile), or obesity (≥95th percentile) were determined. BMI classification for mothers’ was carried out according to the WHO (World Health Organization) guidelines: underweight (<18.5 kg/m^2^), normal weight (between 18.5–24.99 kg/m^2^), overweight (between 25–29.99 kg/m^2^), and obesity (≥30 kg/m^2^) [30].

### 2.3. Statistical Analysis

Differences between the groups were compared using the χ2 test or McNemar’s test where appropriate. The significance of the relationship between the results of parents and children’s taste tests was assessed using the chi-square independence test (Table 2). The chi-square test of independence was used to assess the significance of differences between the incidence of individual BMI categories in different groups (Tables 3 and 4). BMI for this model was defined as obese vs. normal/overweight. The McNemar test was used to assess the differences between the taste preferences of children and their mothers, and thus for two dichotomous features measured on this same group (Table 1). The independence test was carried out each time in a dichotomous division (e.g., obese vs. normal/overweight or overweight/obese vs. normal). We also calculated odds ratios (OR) with 95% confidence intervals for associations between taste preferences and body mass category. Statistical significance was established as a *p*-value of less than 0.05. Calculations were performed with Statistica 10.0 software (StatSoft, Inc., Tulsa, OK, USA).

### 2.4. Ethics

The study was conducted in accordance with ethical standards laid down in an appropriate version of the Declaration of Helsinki and Polish national regulations. The study was approved by the institutional Bioethics Committee at the University of Rzeszow on 02.06.2015 (Resolution No. 15 June 2015) and by all appropriate administrative bodies.

## 3. Results

Seventyboys (47%) and 80 girls (53%) aged 8–15 participated in the study. In this group, 35.3% of children were aged 8–11 years, 64.7% of children were aged 12–15 years. Half of them(44 girls and 31 boys) were overweight or obese. Mothers of children and adolescents participating in the study were aged 27–48. Among them, 69 had excessive body weight (23 were obese). Due to the fact that the number of obese children was too small, no separate analysis was conducted for this group.

No differences in fat taste (character) preference between the children and parents’ were found (*p* = 0.532), while sweet taste preference was significantly more frequent among children than among their mothers (Table 1).

No significant relationship between the preference of fat taste among parents and children was found. A trend towards an association between the sweet taste preference in mothers and children was observed, but it did not reach a limit of statistical significance (Table 2).

Children preferring a high-sweet taste had a 2-fold higher probability of being overweight or obese compared to their peers preferring the only slightly sweet taste. There was no significant relationship between fat taste (character) preferences and the risk of being overweight and/or obese in children. Maternal fat taste (character) preferences were not associated with overweight and/or obesity. Sweet taste preferences were not significantly different between mothers (Table 3). However, 23 mothers were obese (BMI ≥30 kg/m^2^) and 19 of them preferred a high-sweet taste. Such preference, in univariate analysis, was associated with a significantly higher prevalence of obesity, OR = 4.39 (1.71–11.22).

The excessive body weight in children was highly significantly associated with the prevalence of overweight or obesity in mothers, and children of obese mothers had an almost 13-fold higher probability of having excessive body weight compared to children of mothers with normal weight (Table 4).

No significant relationship was found between the mothers’ sweet taste preference and overweight and/or obesity in their children. The relationship between fat taste preferences and overweight and obesity in children, although apparent, it did not reach the limit of statistical significance (Table 4). However, maternal fat taste (character) preferences were significantly associated with obesity in their offsprings (*p* = 0.039) (Figure 2).

## 4. Discussion

In this study, real food products were used to assess taste preferences: apple juice and crackers, which, as some researchers emphasize [34], better reflects the children’s everyday behavior. This is all the more justified as the preferences for sugar content measured in laboratory conditions coincided with the preferences of sweet products such as cereals, beverages [35], or puddings [36]. It was noticed that children and adolescents who were classified in the group with overweight and/or obesity preferred a sweet taste because they were more likely to choose apple juice with higher sugar content. This result confirms the relationship observed in the IDEFICS study, which involved children aged 6–9 from all over Europe, where it was found that among those who preferred a sweet taste, the probability of being obese is 50% higher than in the group that chose a less sweet taste [37].

In this study, there were no significant relationships between the preference for high-fat taste (character) and overweight and/or obesity of children. The analysis of the relationship between the fat taste (character) preferences and the parents’ body weight disturbances also shows no significant correlations, although some literature sources have suggested it [38]. Similarly to the study by Ettinger et al. [39], a mother’s sweet taste preferences were related to overweight and obesity, and in a particularly pronounced way with obesity. This result is in line with the observations of Matsushita et al. [40], who conducted tests on over 29,000 people and found that sweet taste preferences are positively correlated with overweight and/or obesity of women and men.

The study confirmed stronger sweet taste preferences in children. In the case of apple juice tested, half of the children indicated the sweeter one; among mothers, the percentage of indications for sweet juice was lower (35%). It should be added that in the case of tasting juices, the majority of respondents knew which samples had a stronger sweet taste and which were weaker, so their choice was fully conscious. The same conclusions regarding preferences of sweet taste in children aged 5–10 are presented by Mennella et al. [36], who used puddings in her research. Children preferred higher concentrations of sugar in puddings and water solutions. Coldwell et al. [41] believe that during the development, the effect of “desire” for sweets is stronger than the impact of the genotype on the perception of taste. In the same study, it was observed that children chose puddings with a higher fat content less willingly than parents. In our study, no differences were noted in the preference for crackers with different fat content. There is no doubt that children and adolescents are influenced by both good and bad, so habits and nutritional behaviors of parents are very important for shaping taste preferences. It was found that children, with the example of their peers or parents, are more likely to try and accept foods that have not been tolerated before [42].

In our study, no significant relationship between the mother’s taste preferences and children’s taste preferences was found, although in the sweet taste evaluation, some trends were close to the level of statistical significance. Among children of mothers who preferred a sweet taste, about 60% showed the same preferences. In the case of fat taste (character), no relationship was noticed; it can be concluded that the preferences of children do not coincide with the preferences of parents (mothers) and so the direct selection of food products may differ between them. As is commonly accepted, individual, unique preferences and aversions are biologically determined but can be cultivated and modified during child development. Relationships between taste, eating habits, and bodyweight of children may differ from adults due to their limited cognitive abilities, as well as the influence of parents on the quality of children’s nutrition. Additionally, childhood is a key period in shaping the taste, preferences, and eating habits and potential problems with overweight and/or obesity. In a world abundant in foods rich in sugar or its derivatives, sweetened beverages, and processed foods for young children [43], growing children’s heightened preferences for sweets can make them susceptible to excessive sugar consumption [44]. The role of early nutritional experience and its impact on the risk of overweight and obesity of children is well documented [45,46]. Breastfeeding, early exposure to a wide range of products varied in taste, determine later preferences and dietary habits [17,47]. The phase of introducing a complementary feeding is the most important period of learning the taste preferences and the control of appetite in human life. Infants discover sensory impressions (texture, taste, smell) and nutritional properties (energy density) of food that is part of the diet of adults [48]. The hitherto scientific findings suggest that excessive exposure to the sensations of sweet taste can maintain and even strengthen the preference for food products rich in sugars [49]. Children’s preference for high-fat products is explained by association with energy provided by fat. It was observed that higher fat intake at two years of age was associated with lower body weight and leptin in adulthood [50]. The consequence of food intake might be a positive feeling of well-being and satiation or nausea and vomiting associated with unpleasantness [51,52]. In the sensory learning process, these positive or negative experiences are related to the taste preferences of the food consumed, influencing our likes or dislikes.

The obtained results showed the dependence of the fat taste (character) preferences in mothers and the classification of children’s body weight. It is not a very strong dependence; perhaps it should be explained by the influence of preferences—the liking of meals with high-fat content by parents on the type and manner of preparing meals that are prepared for the whole family (including children). The preferences for the sweet taste in parentsare slightly less associated withthe prevalence of overweight in children. Nevertheless, the role of parents in shaping the taste preferences among children and adolescents is very important. Their participation in the development of sensory impressions is not limited to “giving an example” but can have a much wider context. Parents can consciously or unconsciously control the availability of food products and thus influence the exposure to different flavors. In addition, they influence their child’s social environment through their decisions and can indirectly contribute to shaping their sensory preferences [53].

The study results confirm that there is a relationship between taste preferences and overweight and/or obesity in children and adolescents. The taste preferences can also be one of the important factors that determine the formation of overweight and/or obesity in young people. Although there are doubts about whether this is not a reverse relationship, i.e., taste preferences are the result of obesity. In everyday contact with food, the most pleasant is the intensity of the stimulus to which the consumer is accustomed when eating a given product. Possible metabolic dysfunctions that are associated with overweight and/or obesity can lead to changes in hormonal regulation of taste and, as a consequence, changes in taste preferences. In addition, parents of obese children are often restrictive when it comes to eating fat and sugar-rich foods, which can lead to an increase in sweet and fat taste (character) preferences [54,55].

Undoubtedly, the strength of the method used is to imitate the behavior of children in everyday life. The tests carried out were fast and reliable because they were conducted on real food samples known to participants. The present study has several limitations that require further research. First of all, the number of participants in the study was too small to divide it into groups of younger (8–11 years) and older (12–15 years) or overweight and obese. Secondly, it should be noted that other factors like socio-cognitive determinants (e.g., parenting styles, the availability of healthy food at home) were not included in the study. It would be valuable for the results of the study to include, in addition to mothers of participants, the fathers as well.

## 5. Conclusions

The relationships shown in our and some other studies on sweet and fatty taste preferences may affect the formation of inappropriate eating habits in children and adults, and consequently, increase the risk of overweight and obesity, so it is very important to continue research in this area [56]. Primarily, children of parents who are overweight and/or have obesity should be included in preventive measures because the likelihood that they will also face this problem is significant. Adequate family eating habits and the availability of “healthy” food at home lower preferences for foods rich in sugar and fat [57].

## Figures and Tables

**Figure 1 ijerph-17-00538-f001:**
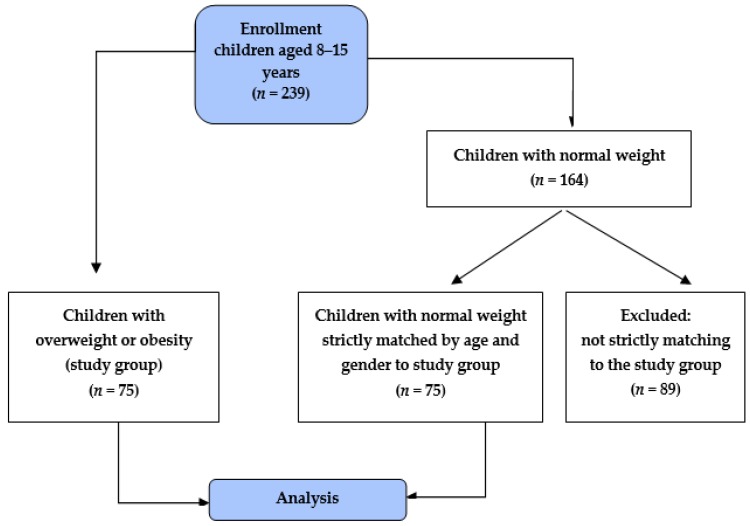
Flow chart demonstrating the selection of the study participants.

**Figure 2 ijerph-17-00538-f002:**
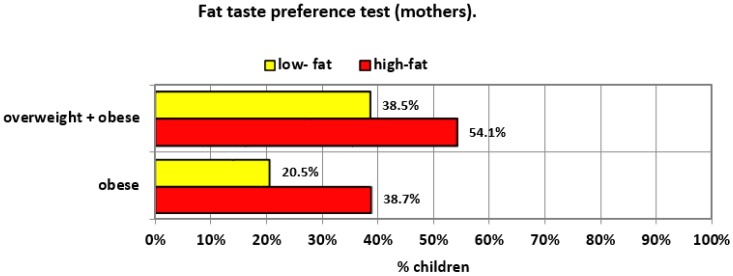
Preferences of the fat taste (character) among mothers and the occurrence of overweight and obesity among children.

**Table 1 ijerph-17-00538-t001:** Preferences of the fat (character) and sweet taste among children and their mothers.

Study Group	Taste Preference Test **			
	Low-Fat	High-Fat	Low-Sweet	High-Sweet
children	45 (30.0%)	105 (70.0%)	75 (50.0%)	75 (50.0%)
mothers	39 (26.0%)	111 (76.0%)	97 (64.7%)	53 (35.3%)
*p*	0.532		0.009 *	

* indicates significantdifference (*p* < 0.05), ** data are expressed as n (%).

**Table 2 ijerph-17-00538-t002:** Relationship between mothers’ and children’s taste preferences.

Taste Preferencein Children	Taste Preference in Mothers **			
	Low-Fat	High-Fat	Low-Sweet	High-Sweet
low-fat	10 (25.6%)	35 (31.5%)	−	−
high-fat	29 (74.4%)	76 (68.5%)	−	−
low-sweet	−	−	54 (55.7%)	21 (39.6%)
high-sweet	−	−	43 (44.3%)	32 (60.4%)
*p*-value	0.4899		0.0603	

** data are expressed as n (%).

**Table 3 ijerph-17-00538-t003:** Probability of being overweight/obese associated with high-fat and high-sweet taste preferences among children and their mothers.

Taste Preference Test	Body Mass Classification				Odds Ratio (95% Confidence Interval)	*p*-Value
	Normal (Referent)		Overweight + Obese			
	N	%	N	%		
low-fat (c)	27	36.0%	18	24.0%	1.78 (0.88–3.62)	0.109
high-fat (c)	48	64.0%	57	76.0%		
low-sweet (c)	44	58.7%	31	41.3%	2.02 (1.05–3.86)	0.034 *
high-sweet (c)	31	41.3%	44	58.7%		
low-fat (m)	24	29.6%	15	21.7%	1.52 (0.72–3.19)	0.272
high-fat (m)	57	70.4%	54	78.3%		
low-sweet (m)	57	70.4%	40	58.0%	1.72 (0.88–3.38)	0.113
high-sweet (m)	24	29.6%	29	42.0%		

(c)—children, (m)—mothers, * statisticallysignificant.

**Table 4 ijerph-17-00538-t004:** Probability of offspring’s overweight and obesity according to the mother’s BMI and taste preference.

BMI Class (Children)	Taste Preference Test (Mother)						Mothers’ BMI Class				
	Low-Fat (Referent) N (%)	High-Fat N (%)	OR (95% CI)	Low-Sweet (Referent) N (%)	High-Sweet N (%)	OR (95% CI)	Normal (Referent) N (%)	Over-Weight N (%)	OR (95% CI)	Obese N (%)	OR (95% CI)
normal	24 (61.5%)	51 (45.9%)	1.88 (0.89–3.97)	53 (54.6%)	22 (41.5%)	1.70 (0.86–3.34)	59 (72.8%)	12 (26.1%)	7.60 (3.35–17.26)	4 (17.4%)	12.74 (3.90–41.63)
overweight and obesity	15 (38.5%)	60 (54.1%)		44 (45.4%)	31 (58.5%)		22 (27.2%)	34 (73.9%)		19 (82.6%)	
*p*-value	0.094			0.124			<0.001 *			<0.001 *	

OR—odds ratio, CI—confidence interval, * statistically significant.

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
