# Peer review of "Preferences for Sweet and Fatty Taste in Children and Their Mothers in Association with Weight Status"

_ijerph, 2020, doi:10.3390/ijerph17020538_

Round 1

Reviewer 1 Report

The manuscript has undergone a substantial improvement from the original submission. However, there are a number of remaining issues.

First of all, other potentially important socio-cognitive determinants of taste preferences NOT assessed in this study should be discussed as study limitations, not just as a passing comment in the response to reviewers.

Secondly, I find the authors' response to my comment about the compound effect of both high fat and high sweet preferences on Ow/Ob to be somewhat inadequate. The other study the authors have referred to is yet to be published and thus there is no evidence of this research. Also, how does this other study stop the analysis of combined sweet+fat associations in the current research? If a combined effect is indeed detected then this would provide additional support for the other study. if the contrary, then perhaps the researchers will need to think about and explain the results were different.

Thirdly, the authors' response to Reviewer 2's comment about sample size calculation doesn't actually answer their question related to study power.

Fourthly, the authors need to address the substantial inconsistencies in formatting between the various tables. The biggest inconsistency that I can see is the use of "high-fat/sweet" and "low-fat/sweet" in some tables, and the use of "non-fat/sweet" and "fat/sweet" in others. Please address other inconsistencies.

Figure 1 also needs to be amended to remove all the carriage returns and other formatting symbols.

Finally, the authors have misinterpreted my comments about the typos in the 2.1 Participants section. "AN randomized algorithm" isn't grammatically correct. Same goes for "69 of them WERE overweight and/or obesity..."

Reviewer 2 Report

Methods

state whether the sample has representative

provide the calculation of sample size

describe whether the sample selection was selected the child-parent pair, what is the inclusion, exclusion criteria?

describe the body weight meaurement procedure, whether the study participant in fasting status? How the the quality control for the antropometry were carried out?

Result:

present the general informtion of the study participants

Reviewer 3 Report

In the revision, the authors have clarified that the administered samples were blinded, thus addressing the most important methodological concern that I had. The authors have also addressed my other comments. A few concerns still remain, mainly about the readability of the results tables. The authors should consult a knowledgeable colleague experienced in preparing tables of results for publication.

The added text in section 2.3 (stat methods) partially addresses my prior comment, however no details were provided on the model upon which the odds ratios are based. Was it logistic regression? Please state so. Was BMI for this model defined as obese vs. normal/overweight? Please state so in the text. Also, the added text in this section needs to be proof-read for grammar. Several blank spaces between words are missing. The reformatted tables now look even worse than before and are totally unreadable. Table 2 is just two tables stacked together, with their separate column headings, which does not make sense. Make the unified column headings for the entire table, and separate sweet or fat test results using groups of rows. Please consult a knowledgeable colleague or published papers in any journal to see how this is typically done. Table 3 is also unreadable. Please transpose Table 3: use column headings “Test preference”, “Normal”, “Overweight+Obese”, “Obese”. Use row groups for sweet and fat test preference. Table 4 is also unreadable. Reformat Table 4 using uniform column headings across the table, similar to Tables 2 and 3 (see comments above).

Author Response

This manuscript is a resubmission of an earlier submission. The following is a list of the peer review reports and author responses from that submission.

Round 1

Reviewer 1 Report

General comments:

The study by Sobek et al describes differences in sweet and fat taste preferences in children and mothers of varying weight status. This is an interesting area of research, but as the authors have already alluded to in their conclusion, "sweet and fatty taste preferences MAY affect the formation of inappropriate eating habits". Taste preference is only one of many socio-cognitive determinants of children's eating habits. I wondered if the researchers of this study collected any data on other influencing/confounding factors e.g. parenting styles, availability of healthy food at home etc? It would be important to consider how these other factors contribute and interact with taste preferences to predict child overweight and/or obesity.

In addition to this, did the authors consider looking at how sweet and fatty taste preferences interacted to influence overweight/obesity? For example, did having both a high-sweet and high-fat taste preference increase the odds of having overweight and obesity? And was there also a dose response relationship in that children with both high fat and sweet preference had greater severity of obesity than those who only had a high-sweet preference?

Specific comments:

Page 2 line 3 – avoid using double negative here with “did not bring unambiguous conclusions”. It just makes reading more difficult.

Page 2 (section regarding the IDEFICS in the introduction) – The reason you have mentioned this is because you have adopted their methodology in your current study. However, the way this sentence has been written at present sounds like you are about to present some background knowledge and findings from the IDEFICS studies, which is not the case. If there are findings from these studies relevant to the current paper then please expand, otherwise please reword so that readers are not expecting to hear about results of these studies.

Page 2 section 2.1 line 3 – AN randomized algorithm

Page 2 section 2.1 line 11 – WERE overweight and/or obesity

Table 2 – Please avoid the use of “effect” as this implies causality and this is only a study of association.

Discussion page 6 – The authors touch of results of maternal and child taste preference correlations. These data should be presented more systematically in the results, and not appear as a passing comment in the Discussion.

Reviewer 2 Report

1、please describe what the study design? what is the assumption?

2、how the sample size was assessed and calculated, on what basis of?

3、describe the inclusion and exclusion criteria for the participants.

4、for the weight measurement, if the participants are in fasting status, whether the anthropometric measurements following a standarized procedure?

5、The presentation of the results should be rewritten. The characteristics should be described. The general preferences should be described by age and gender. Tables should be considered combined together.

6、The conclusion is a bit too long, need to be revised.

Reviewer 3 Report

The paper reports on the study of children’s and their mothers’ taste preferences in relation to their weight status. The idea of the study is interesting, and would make for an interesting report if properly executed. However, there are serious methodological flaws that bring the validity of results into question and possibly completely invalidate the study. Following are my specific comments.

Major comments:

Section 2.2: How many pairs of juices and/or crackers were given to each study participant? If only one, and the participant was forced to choose, the choice could have been random (if no preference). Thus, assessment of taste preferences based on one sample pair is prone to very high measurement error and therefore highly questionable. In this case, the results of the entire study should be deemed unreliable. Please clarify. (Note: this comment is only applicable if blinded samples are administered, which apparently was not the case - see below.) Discussion: “It should be added that in the case of tasting juices, the majority of respondents knew which samples had a stronger sweet taste and which were weaker, so their choice was fully conscious.” This piece of info belongs in the Methods, as it completely changes the entire study design and interpretation of results. The absence of blinding calls into question the validity of taste measurement and of the results. I would argue that due to the absence of blinding, the obtained taste preference measurements do not necessarily reflect the true taste preferences, but the engraved psychological perceptions about sweetness and/or fatness: “I decide that I like sweet taste, therefore I will consciously choose sweet juice.” Please note that in this case, real food samples are unnecessary – the same could have been accomplished just by asking the participants whether they prefer sweet and/or fat taste. Thus, the absence of blinding did not allow for a reliable assessment of true taste preferences, and completely invalidates the study. The only way this paper can be saved if it is rewritten in terms of psychological perceptions and/or preferences.

Minor comments:

Introduction:

Line 4: the statement that “the choice of food consumed by children depends mostly on their taste preference” is debatable: this choice of food may in large part depend on their parents’ choices. Please provide a reference to support this statement. Intro, last paragraph: the significance of studying parents’ taste preferences in relation to children’s weight status has not been established. Please add to the Intro to establish such significance. Intro, last paragraph: please end the Intro section with a brief summary of the present study’s goals and study population/location.

Section 2.1:

Please state the study location (city/country). The ‘strict matching’ on age is unclear. What is the ‘the nearest date of birth’? Were controls’s age within 1 year of the matched study group participants? The sentence “69 of them were overweight and/or obesity” belongs in the Results.

Section 2.3:

The description of stat methods is totally insufficient, offering almost no details. Please specify when exactly the chi-square test was used, and when McNemar’s test was used. Please also specify what model the odds ratios were based on, what was the outcome measure in that model (and how it was coded), and state all the covariates in that model. If the outcome was BMI, was it coded as overweight/obese (yes/no), or normal/overweight/obese with a polytomous model? (Note: it is more or less clear from Table 2, but should be explicitly stated in the Methods.)

Results:

There are too many tables, which are very small. Please combine tables 2, 3, 4, and 5, perhaps transposing them for better readability. Table 4 is missing normal-weight counts. Table 5 is missing normal-weight counts and obese only counts and OR. Add to Table 6 the results of the model testing the association of mothers’ taste preferences (sweet and fat) with their children’s weight status. Figure 1 does not contain enough info on this model: counts and OR (CI) are missing and mothers’ sweet preference vs. children’s weight status is missing. Transpose Table 6 for better readability.